
Comparison of two nationwide lightning location systems and
characteristics of could-to-ground lightning in China
**Ruijiao Jiang[a], Guoping Zhang[a,*], Shudong Wang[a], Bing Xue[a], Zhengshuai Xie[b],**
**Tingzhao Yu[a], Kuoyin Wang[a], Jin Ding[a], Xiaoxiang Zhu[a]**
[a] Public Meteorological Service Center, China Meteorological Administration, Beijing 100081, China
[b] Weather Modification Centre, China Meteorological Administration, Beijing 100081, China
* Corresponding author. E-mail address: zhanggp@cma.gov.cn
Abstract

9       The lightning location system based on multiple sub-stations is an effective means

of lightning observation. This study first compares the two nationwide lightning
location systems in China, the Advanced TOA and Detection System (ADTD) and the
Three-Dimensional Lightning Location System (3D-LLS), using the observations in
2020. As a significantly updated version of ADTD, 3D-LLS has a cloud-to-ground (CG)
detection efficiency of nearly twice ADTD. However, its scarce distribution sites in
central Tibet, western Xinjiang, eastern Qinghai, and eastern Heilongjiang account for
some blind observation areas where ADTD could play a supplementary role. The ratio
of +CG and the distribution of currents indicate that 3D-LLS misjudges a certain
number of intracloud (IC) pulses as return strokes. Besides, the IC detection developed
by 3D-LLS is still inefficient. In the results, the IC flashes only account for half of CG
flashes, much smaller than the regular ratio (twice). According to the comparison with
the nearly no-miss optical observations in Guangzhou, the CG detection efficiency of
ADTD and 3D-LLS herein is 24.5% and 50.5%. Further improvements are expected in
terms of the above shortages. Still, the ground-based observation has better detection
efficiency and accuracy than the satellite, especially for CG lightning. The dataset from
ADTD in 2016-2021 is employed to analyze the lightning characteristics' temporal and
spatial distributions and the difference between +CG and -CG over China. It can be
concluded that low latitude, undulating terrain, seaside, and humid surface are favorable
factors for lightning occurrence. Thus the southeast coastland has the largest lightning
density, while the northwest deserts and basins and the western and northern Tibetan
Plateau have almost no lightning. For the period with high CG frequency (summer of a
year or afternoon of a day), the ratio of +CG and the discharge intensity is relatively
small. The Tibetan Plateau leads to the complexity of lightning activity in China and
lays the foundation for studying the impact of surface elevation on lightning. Results
indicate that the +CG ratio on the eastern and southern plateau is up to 15%, and the
west and north sides have a low percentage. The discharge intensity of +CG and -CG
on the Tibetan Plateau is approximate, while the +CG always has a larger current than





-CG on the plains.
**Keywords**:China, ADTD, 3D-LLS, Comparison, Lightning characteristics, +CG

# 1.Introduction

Most lightning is generated mainly through meso-small scale severe convective
weather, with few occurring in stratus clouds and tropical cyclones and occasionally
during volcanic eruptions, nuclear explosions, and dust storms (Rakov and Uman,
2003). Lightning, a violent long-distance transient discharge phenomenon, could cause
severe disasters such as human and animal casualties, forest fires, and electronic and
communication equipment interruptions. Therefore advanced lightning monitoring
technology is necessary for the development of lightning science and also scientific
protection against lightning disasters.
Lightning discharge emits electromagnetic spectrums with a broad range,
providing an essential avenue for lightning detection. The very low frequency / low
frequency (VLF/LF, 20-300 kHz) band radiation is mainly produced by the cloud-to-
ground (CG) return strokes, intracloud (IC) K-changes, and other discharge processes
with a large spatial scale. VLF/LF electromagnetic waves could propagate along the
ground surface or be reflected between the surface and ionosphere propagation, with
superiority of long propagation distance (hundreds to thousands of kilometers) and slow
attenuation. This frequency range thus is suitable for large-scale lightning detection and
is currently the most commonly used target detection band for ground-based lightning
location systems.
Representative examples of modern lightning location systems working in
VLF/LF band are mainly the U.S. National Lightning Detection Network (NLDN), Los
Alamos Sferic Array (LASA), European Cooperation for Lightning Detection
(EUCLID), etc. The three nationwide detection networks in China are the Advanced
TOA and Detection System (ADTD) operated by the Meteorological Observation
Centre of China Meteorological Administration (CMA), the Lightning Location System
(LLS) of the State Grid Corporation of China, and the Three-Dimensional Lightning
Location System (3D-LLS) deployed by the Institute of Electrical Engineering of
Chinese Academy of Sciences (CAS). There are also small-scale and refined detection
systems in local areas, such as the Beijing Lightning Network (BLNET) established by
the Institute of Atmospheric Physics of CAS, the Guangdong-Hongkong-Macao
Lightning Location System (GHMLLS) deployed by the meteorological departments
of Guangdong Province, Hongkong, and Macao.
Due to the differences in equipment types, instrument errors, calculation principles,
and installation environments, there are certain deviations in the detection results of
different lightning location systems. To improve the reliability, effective utilization rate,
and practical application effect of the systems, and also to provide a theoretical basis
for the optimization of future networking efficiency and the fusion of multi-source



lightning location data, it is crucial to evaluate and analyze the quality of the networks.
Currently, there are two dominant methods for assessing lightning location systems.
One is to use a limited number of rocket-triggered lightning cases and lightning current
measurement data or optical image data of high structures as references to statistically
evaluate the current or location deviation (Jerauld et al., 2005; Nag et al., 2011; Schulz
et al., 2016). Another way is the comparison between different systems (Bitzer and
Burchfield, 2016; Murphy and Said, 2020; Srivastava et al., 2017). The former gives
more accurate assessment results but is only available when the striking location can be
confirmed by other sources. For vast observation areas, the assessment of detection
accuracy and efficiency can only be obtained by comparing networks, often using one
with known better performance to assess another. If the detection efficiency (DE) of
both systems has not been evaluated before, a Bayesian algorithm can be used to find
the relative DE (Bitzer and Burchfield, 2016; Bitzer et al., 2016).
China spans a wide range of latitudes from north to south and significant terrain
changes from east to west, and the western and northern parts of the Tibetan Plateau
have large uninhabited areas with altitudes above 4500 m. The above factors pose
challenges for the installation of lightning detectors and the improvement of the
accuracy of locating algorithms. Among the three nationwide systems, only ADTD and
3D-LLS are available for us. In reviewing the literature, comparative evaluation of
these two networks is lacking and mainly aimed at localized areas. In this paper, we use
the nationwide CG data in 2020 obtained from ADTD and 3D-LLS to compare their
capabilities in terms of lightning density, time difference, relative positioning accuracy,
and current peak value. The consequences could assess the quality and reliability of the
two datasets and their application in different scenarios. In addition, China's wide
latitude and longitude range and complex topography make for studying the
relationship between lightning and geographic factors. This study makes use of ADTD
data from 2016-2021 to analyze lightning distribution in China and the temporal and
spatial distribution difference between +CG and -CG.
## 2.Comparison of ADTD and 3D-LLS
### 2.1 Introduction to the two networks
ADTD was first developed by the National Space Science Center (NSSC) of CAS
in 2007 and is currently operated by the Meteorological Observation Centre of CMA.
The system consisted of 435 sub-stations (by 2020) equipped with lightning detectors
and the central data processing station deployed at the National Meteorological
Information Center. ADTD can generally achieve nationwide detection without blind
areas and is nowadays the most widely used system by the meteorological departments
in China.
3D-LLS has been in operation by the Institute of Electrical Engineering of CAS
since 2013, which is an upgraded version of the equipment and algorithm of ADTD,



adding the function of detecting IC lightning and improving the efficiency of CG
lightning. 3D-LLS also went on to scale over 400 sub-stations by 2020, but thick
distributed in southern and eastern China and sparsely in western China. Part of the
station distribution in 2018 can be found in Wang et al. (2020). 3D-LLS has also
expanded to other Asian countries such as Sri Lanka, Myanmar, Cambodia, and Korea.
A time-of-arrival (TOA) technique with a GPS timing error of fewer than 20 ns
(clear sky) is used by both two networks, and the detecting targets are the VLF/LF
pulses of CG return strokes (and IC K-changes). A lightning flash might comprise
several CG strokes or IC K-changes, and both systems group single-point signals to a
flash event according to their separation in time and space. However, their rules of
categorization are different, as ADTD classified the return strokes within 500 ms time
interval and 10 km distance interval as a single CG flash, while 3D-LLS set the
thresholds as 1 s and 10-30 km to group a single CG or IC flash and distinguished the
two types by the discharge height. This is one factor leading to the inconsistency in the
number of lightning flashes between the two networks.
In this study, ADTD and 3D-LLS datasets were downloaded from the CMA big
data cloud platform. Time of occurrence, latitude, longitude, current peak value,
number of located stations, (type of lightning) for each flash was obtained.
2.2 Comparison of detection results
Since 3D-LLS had been adding sub-stations until 2020, this research only used the
data in 2020 to compare the performance of the two networks. As 3D-LLS only retained
lightning detected by five or more stations simultaneously this year, accordingly, this
study did the same for ADTD data. Therefore, the retained cases were highly reliable,
but it led to some missing cases where stations are sparse.
In 2020, ADTD detected a total of 5,898 thousand CG flashes in China, in which
+CG flashes accounted for 8.5%. 3D-LLS detected 10,464 thousand CG flashes, with
a DE nearly twice that of ADTD and a +CG flash percentage of 21.4%. In addition, 3D-
LLS also detected 4,250 thousand IC flashes. In general, IC flashes account for 2/3 of
the total number of lightning flashes (Rakov and Uman, 2003). It can be inferred that
the missed rate for IC is greater than CG, as the discharge intensity of IC K-change is
much smaller than that of CG stroke, which is also the technical bottleneck of most
ground-based lightning location systems at present.
In our previous study, we calculated the lightning density in the vicinity of the
Canton Tower in Guangdong using optical observation by the Tall-Object Lightning
Observatory in Guangzhou (TOLOG), which is currently regarded as a nearly no-miss
observation (Wu et al., 2019; Jiang, 2021). The result was that the CG flash density was
20 km$^{-2}$ yr$^{-1}$ within a 3 km radius of the Canton Tower. The detection results in the same
region for ADTD and 3D-LLS are 4.9 km$^{-2}$ yr$^{-1}$ and 10.1 km$^{-2}$ yr$^{-1}$, respectively. The
DE of the two networks correspondingly is only 24.5% and 50.5%, and improvement
is expected.
For further comparison, China's land area is divided into a 0.25°×0.25° grid




according to the latitude and longitude, and the annual CG flash density is calculated
for each grid. Fig. 1(a,b) shows the distribution results of ADTD and 3D-LLS in 2020.
Due to the huge regional differences, an exponential color scale is used here, with
regions of high density indicated by red and regions of low indicated by blue, with a
transition color in between. The overall distribution of the two networks is similar, and
the high-value and low-value regions correspond well. The specific distribution
characteristics will be detailed in section 3.1.

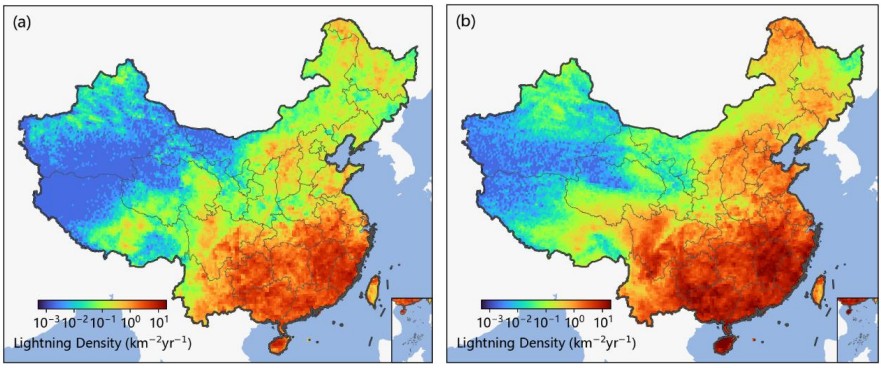

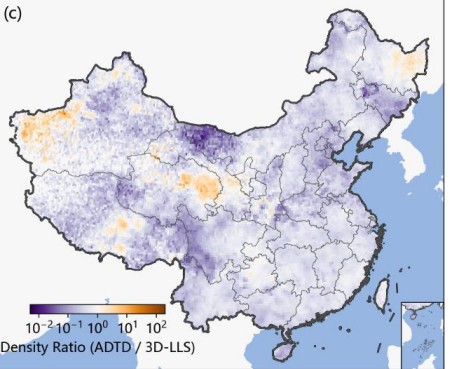

**Fig. 1. Distribution of CG flash in China detected by (a) ADTD and (b) 3D-LLS in 2020; (c) ratio of CG flashes detected by the two networks in 2020 (ADTD/3D-LLS). The grid size is 0.25°×0.25°, and the map is in Lambert's equirectangular cone projection, the same below.**

The distribution of DE difference is shown in Fig. 1(c), calculated by dividing the
grid densities of the two systems (ADTD/3D-LLS). When their DE is similar, it is
represented by white; when the DE of ADTD is higher, it is represented by brown; when
the DE of 3D-LLS is higher, it is represented by purple. The darker color corresponds
to the bigger difference. The DE difference between the two systems can be up to a
hundred times. In general, 3D-LLS has a higher DE as been significantly upgraded, but
its site distribution is relatively uneven. It is particularly dense in some parts of Yunnan,
Sichuan, Tibet, Inner Mongolia, Hebei, Jilin, etc. Thus the DE of these regions is high.
In contrast, the number of stations deployed in central Tibet, western Xinjiang, eastern





Qinghai, eastern Heilongjiang, etc., is scarce, even leading to some blind detection areas.
Overall, the DE of 3D-LLS is about two times that of ADTD.
The current peak value of CG detected by the two networks is compared (removing
outliers above ±300 kA), as shown in Fig. 2, with the orange histogram representing
ADTD and the slash histogram representing 3D-LLS. Most of the lightning discharged
with the current peak between -100 kA and 100 kA. The negative currents measured by
ADTD are generally larger than 3D-LLS, with an average ratio of 1.46. For +CG, the
current peak distributions of ADTD and 3D-LLS are approximately the same when the
value is greater than 30 kA, with a ratio of 1.03. However, 3D-LLS has a large number
of outliers in the 0-30 kA range, which is presumed to result from misclassifying IC as
+CG, leading to a high percentage of +CG of 21.4%. This phenomenon is mostly
because that 3D-LLS distinguished IC and CG by the height of the radiation source.
According to the principle of the TOA algorithm, the positioning error in the vertical
direction is much larger than that in the horizontal direction, so a significant number of
misjudgment cases appeared. Merge the electric field waveform identification to the
discrimination algorithm might help to improve the recognition accuracy. ADTD had
already eliminated the +CG with a current peak of 0-10 kA from the original data to
reduce misjudgment.

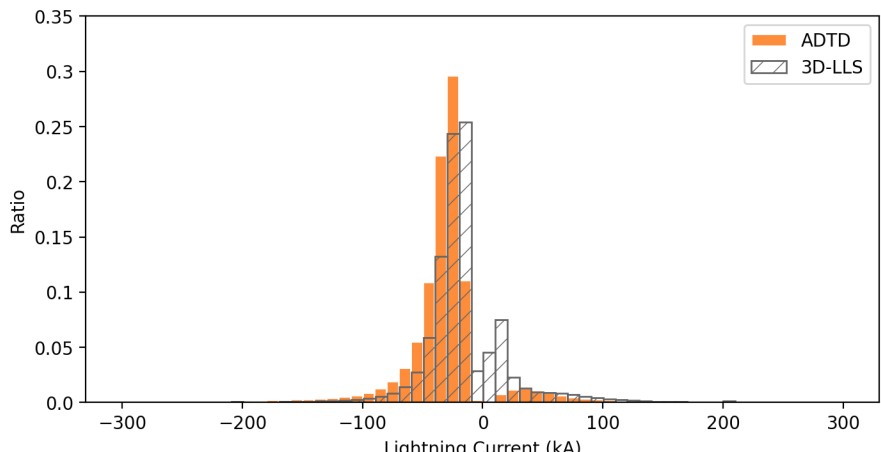


**Fig. 2. Distribution of current peak value of CG flashes detected by the two networks in 2020**
2.3 Match of the same CG flash from two datasets
Comparing the detection results for the same radiation source is necessary for
valuing the difference between the two networks. The spatial and temporal thresholds
for matching vary between studies, with most having a time difference (Δt) threshold
of a few hundred ms and a spatial distance (Δr) threshold of 10-30 km (0.1 ms & 1 km
(Pohjola and Mäkelä, 2013), 1 s & 15 km (Poelman et al., 2013), 200 ms & 20 km
(Bitzer and Burchfield, 2016), etc.). Different matching thresholds bring differences to
the results. In our study, when the Δt threshold is set to be 1 s, 3,219 thousand CG



flashes are matched if the Δr threshold is set to be 30 km, and the match number is
3,079 thousand if the Δr threshold is set to be 10 km. It means the Δr threshold has no
significant effect on the matching results, and the number of matched CG flashes is
about 60% in ADTD and 30% in 3D-LLS. The percentages imply that even though the
DE of 3D-LLS has been improved a lot, a large number of flashes is still undetected.
By expanding the Δt distribution in the 0-1 s interval, it can be found that Δt is evenly
distributed in the 20 μs-1 s interval, while a large proportion of Δt in the 0-20 μs (0.002%
of 1 s) interval, as shown in Fig. 3(a). There are 64 thousand (1.1% of CG flashes by
ADTD, 0.6% of CG flashes by 3D-LLS) flashes with Δt in 0-20 μs, and the
corresponding Δr distribution is shown in Fig. 3(b), with 86.5% within 2 km and 96.8%
within 4 km. It is reasonable to assume that these radiation sources came from the same
lightning strokes recorded by two systems. Bitzer and Burchfield (2016) indicated that
the Δt of the same stroke by different networks should be no more than 50 μs in case
they are timed by GPS. The two networks have different criteria for grouping flashes,
and if there was a missed stroke in a lightning flash and they did not use the same stroke
to represent the flash, the same lightning flash could not be matched in this case, leading
to the low matching ratio.

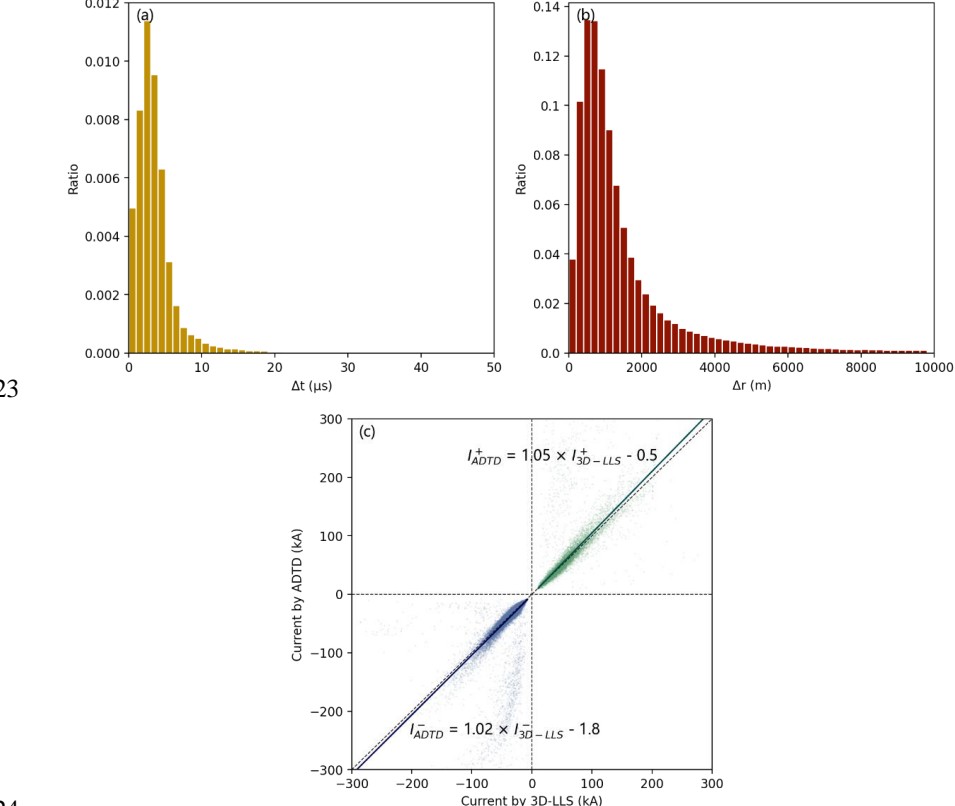






**Fig. 3. Difference between the matched lightning stroke of the two systems. (a) Δt distribution; (b) Δr**

**distribution; (c) current comparison, green represents +CG, and blue represents -CG, same below.**

The same stroke currents detected by the two systems are compared, and +CG and -CG are indicated by different colors, as shown in Fig. 3(c). For most of the strokes, the current measurements are very close. However, there are some anomalies that the ADTD currents are significantly higher than the 3D-LLS currents. By examining the locations of these points, they are concentrated within a 50 km radius of the junction of Guizhou, Guangxi, and Hunan. Since there is no special feature in this area, it is inferred that some ADTD sub-stations might work abnormally. After eliminating these anomalies, the currents of +CG and -CG are fitted, respectively, and the relationship is marked in Fig. 3(c) with $r^2$ of 0.90 and 0.81. The current ratios after matching strokes are more accurate than the generalized current ratios calculated in Section 2.2. By matching and fusing the detection results of the two systems, the advantages can be well complemented, and the existing resources can be fully exploited to achieve further optimization of existing equipment.

# 3.CG characteristics of China

Although the DE of ADTD is less than that of 3D-LLS in the areas with dense site distribution, ADTD has a relatively more uniform site placement and a smaller blind spot range. Moreover, the available years of 3D-LLS are short, and the misjudgment of +CG is common. Therefore, the ADTD dataset from 2016-2021 is utilized to analyze China's lightning characteristics. The data are filtrated by the number of triggered stations (≥3). Positive strokes with small peak currents (<10 kA) are likely to be misclassified as CG discharges when those are more likely to be of intracloud nature (Cummins et al., 1998), so they are discarded from the dataset.

## 3.1 CG distribution in China

Fig. 4 showed the surface height distribution of China based on the global surface elevation data (resolution of 30 m) from NASA's new generation earth observation satellite, Terra. The large latitudinal span, the great terrain disparity, and the complex topography varied the climate features in China. As a fundamental meteorological element, lightning mainly occurs in meso-small scale thunderstorms, and its long-time accumulation characteristics are closely associated with the climate of China. Atmospheric circulation, topography, distance from the sea, latitude, etc., jointly determine the terrestrial spatial distribution of lightning in China.


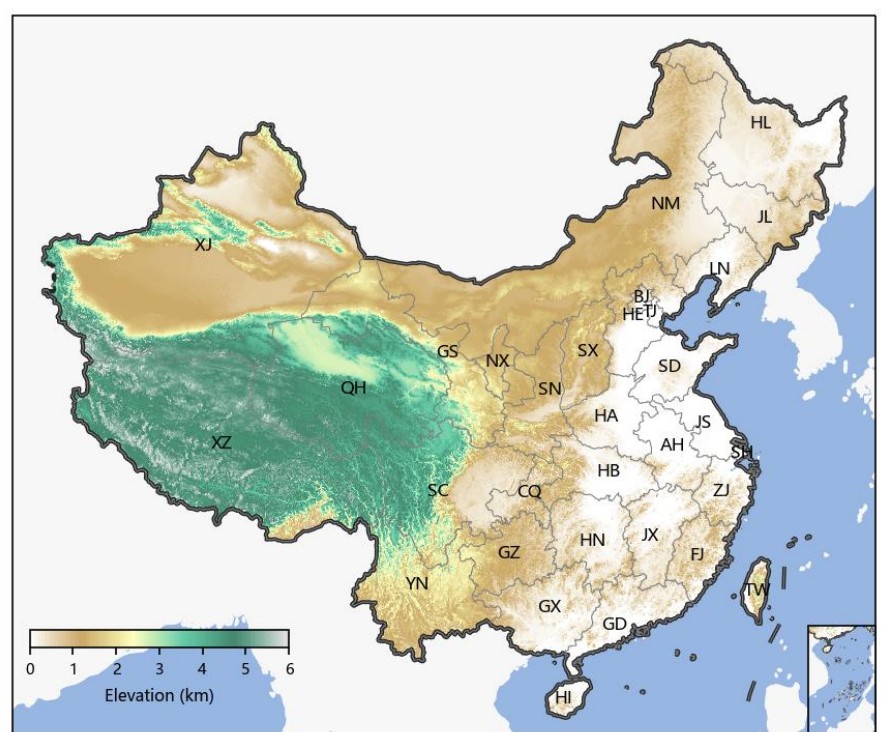

**Fig. 4. Altitude distribution map of China with the location of each province and municipality (indicated by abbreviations, for details, refer to:** *https://www.iso.org/obp/ui/#iso:code:3166:CN***)**

China can be divided into four major geographical regions, namely, southern China (Region-I), northern China (Region-II), northwestern China (Region-III), and the Qinghai-Tibet region of China (Region-IV), as shown in Fig. 5. Qinling Mountains-Huaihe River line (roughly coinciding with the 0 ℃ isotherms and 800 mm annual precipitation line in January) is the dividing line between Region-I and Region-II. The Daxing'an Mountains-Yinshan Mountains-Helan Mountains (dividing monsoon and non-monsoon and the 400 mm annual precipitation line) is the boundary between Region-II and Region-III. The dividing line between Region-IV and Region-I-II-III is roughly the line between China's terrain's first and second steps.

Fig. 5 shows the annual average CG flash density distribution from 2016-2021. Most CG flashes are concentrated in Region-I with a density greater than 1 km$^{-2}$ yr$^{-1}$. The leap line of lightning density corresponds well with the 0 ℃ isotherms in January, the 800 mm annual equivalent precipitation line, and the eastern dividing line of the first and second terrain steps. The climate in Region-I is mainly influenced by the tropical\subtropical monsoon. The southeast monsoon from the Pacific Ocean and the southwest monsoon from the Indian Ocean makes the summer hot and humid and prone to thunderstorms. In particular, the monsoon influence is more pronounced in the coastal areas with abundant water vapor and thermal conditions. In the mountainous



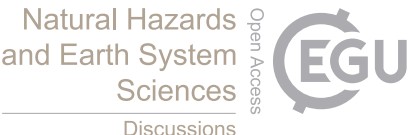
regions of Hainan, Guangdong, Fujian, and Zhejiang, where the rolling topography lifts
the warm and humid air masses, thunderstorm activity is most frequent, so the lightning
density is especially high. Although the Sichuan Basin and Yunnan are far from the
coastline, they are located at the eastern and southern windward slopes of the Tibetan
Plateau, which benefits the generation of thunderstorm activities due to the topographic
uplift.

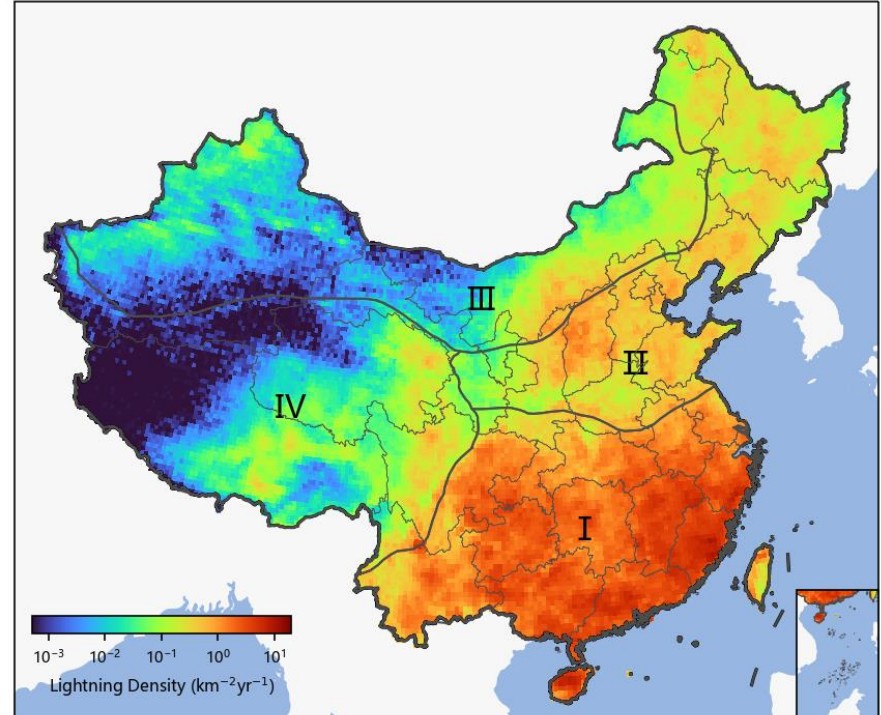


**Fig. 5. 2016-2021 annual average CG flash density distribution in China (Region-I: southern China; Region-**
**II: northern China; Region-III: northwestern China; Region-IV: Qinghai-Tibet region of China)**
Region-II has mainly the temperate monsoon climate, with summer influenced by
the southeast monsoon carrying temperate marine air mass or degenerate tropical
marine air mass, making summer warm and rainy. Most areas have CG flash density
between 0.1-1 $km^{-2}$ $yr^{-1}$, slightly lower than Region-I. The lightning density in Region-
II is also greater in the seaside area than inland areas. Shanxi is located in mountainous
region, and the undulating terrain makes it a high incidence area for thunderstorm
activity. Region-II has the most extensive plain, Northeastern China Plain, which is in
the shape of a trumpet, surrounded by the Daxing'an Mountains-Xiaoxing'an
Mountains-Changbai Mountains. The landform is conducive to the southeast monsoon
to reach the inland areas of Region-II and form summer thunderstorms. Jilin is only a
dozen kilometers from the Sea of Japan, facilitating the entry of Japanese warm air




currents. Therefore, thunderstorm activity is relatively intense in Region-II despite its
high latitude.
Region-III, including Xinjiang, northern Gansu, and most of Inner Mongolia, has
a temperate continental climate. The southern and central parts of Region-III are mostly
vast deserts and gobies. The Tibetan Plateau blocks the humid South Asian monsoon,
and its arid surface cannot produce abundant water vapor, so almost no thunderstorms
are generated here. Although the Tianshan Mountains, Kunlun Mountains, Altay
Mountains, and Tarbahatai Mountains in Region-III are located in the hinterland of the
Eurasian continent, they are still provided with water vapor for thunderstorm generation
from the westerly circulation transporting evaporated water vapor from the Atlantic
Ocean and the Eurasian continent, making the northern mountainous areas occupy
almost all the lightning activity in Xinjiang. The southeastern monsoon flowing through
Region-II, reaching the eastern and central mountainous regions in Inner Mongolia in
summer, can still bring some thunderstorm processes.
The main body of Region-IV is the Tibetan Plateau, mainly including Tibet,
Qinghai, southern Xinjiang, and western Sichuan. It has a highland mountain climate,
and the overall geomorphic distribution trend is increasing from west to east (Ma et al.,
2021). The uninhabited areas above 4500 m in elevation in the west and north of
Region-IV are icy all year round, covered by snows and glaciers. The Qaidam Basin in
Qinghai is a closed, huge interrupted basin, with dry sinking airflow from the northern
edge of the plateau in summer, leading to water shortage. There are few thunderstorm
activities in the areas mentioned above, and the distribution of sub-stations is sparse,
making them the regions with the lowest lightning density detected in China, with CG
flash density less than $10^{-3}$ km$^{-2}$ yr$^{-1}$. The eastern Himalayas, near the Yarlung Tsangpo
River Grand Canyon, has relatively low altitude, opening a "gap" for the influx of
abundant water vapor from the Bay of Bengal. The rapid climb of water vapor leads to
frequent thunderstorms. The thunderstorms around Nagqu in the central part of the
plateau are mainly located between the east-west Himalayas Mountains and Tanggula
Mountains. The thunderstorms on the east side of the plateau are mainly influenced by
the low vortex and the shear line, which is usually stable at around 32.5 °N. The high
lightning density area is located precisely on the south side of the shear line.
The annual average CG flash density of each province and municipality is
calculated and ranked in Fig. 6, and their geographical locations are shown in Fig. 4.
Among them, Fujian (FJ) has the leading average CG flash density with more than 4.5
km$^{-2}$ yr$^{-1}$, followed by Hainan (HI), Zhejiang (ZJ), Jiangxi (JX) and Guangdong (GD)
with more than 3 km$^{-2}$ yr$^{-1}$. Guangxi (GX) and Guizhou (GZ) have a density of around
2.5 km$^{-2}$ yr$^{-1}$, while other inland provinces in Region-I are less than 2 km$^{-2}$ yr$^{-1}$. The
density of Shanxi (SX) in Region-II is the highest, close to 1 km$^{-2}$ yr$^{-1}$. In the other
provinces and municipalities in Region-II, Henan (HA), Shaanxi (SN), Hebei (HE),
Beijing (BJ), Shandong (SD), Tianjin (TJ), Heilongjiang (HL), and Liaoning (LN) have
the close value between 0.2-0.4 km$^{-2}$ yr$^{-1}$. The density in Ningxia (NX), Qinghai (QH),
Gansu (GS), Tibet (XZ), and Xinjiang (XJ) in Region-III and Region-IV is the lowest

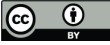
in China, which is less than 0.1 km$^{-2}$ yr$^{-1}$.

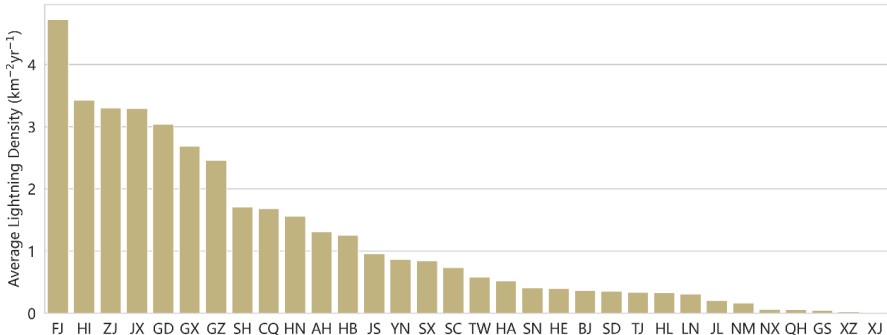

**Fig. 6. Comparison and ranking of annual average CG density by province and municipality**
3.2 Differences between +CG and -CG
According to the different polarities of neutralized charge in thunderclouds, CG
can be divided into +CG and -CG. Compared with -CG, +CG generally has a lower
probability of occurrence, only about 10%, but has a larger spatial scale and charge
transfer, resulting in a more robust hazard (Preston and Tolver, 1989). Much work has
been done on the comparative study of +CG and -CG in local areas (Nag et al., 2014;
Rakov and V., 2003), and on their basis, this study further analyzes the spatial and
temporal variability of +CG and -CG in China with the complex climatic and
geographical factors.
3.2.1 Comparison of the temporal distribution of +CG and -CG
Fig. 7(a) shows the monthly average CG flash frequency distribution for six years
from 2016-2021 in China. The frequency fluctuates significantly in different months,
with the most frequent occurrence in August (-CG flashes up to 1,345 thousand and
+CG flashes up to 152 thousand). December has the least number, with only 5 thousand
-CG flashes and less than 400 +CG flashes. Lightning activity is also rare in November,
January, and February, with an abrupt jump in March and a transitional increase in
subsequent months. According to the seasonal classification, lightning activity is most
active in summer (June, July, and August), accounting for 70.7% of the year. In other
seasons, lightning is more frequent in spring (19.1%) than in autumn (9.8%) but much
less than in summer. The main reason is that the summer monsoon affecting China has
started to form during April and May, while the cold and dry winter monsoon starts to
build up and push southward from September, making thunderstorm activity in spring
and autumn mainly concentrated in southern areas, especially coastal areas. In winter,
China's mainland is controlled primarily by cold high pressure, when there is very little
lightning in most regions and only a small amount of lightning occurs in the
southeastern coastal areas, accounting for only 0.4% of all year round. Overall, the
seasonal trend is that the lightning distribution advances from south to north and then



retreats southward, which is consistent with the trend of the summer monsoon. In
addition, the proportion of +CG flashes in different months is calculated, indicated by
the gray line in Fig. 7(a). The proportion of +CG flashes and the lightning frequency
have an obvious inverse trend. The proportion of +CG is low, less than 10%, in the
months with frequent lightning. In contrast, the ratio is very high in winter, when
thunderstorms are not easily generated, even up to more than 40% in December.
The current peak value of the two types of CG flashes in different months is
analyzed, as shown in Fig. 7(b). On the whole, the distribution range is wider in winter
than in other seasons. The average of each month is shown by the white cross in the
figure and the variation trends by red lines. From the results, the discharge intensity of
+CG flashes is greater than -CG flashes. The current of +CG and -CG flashes is the
largest in January and December. The difference between -CG flashes is not obvious in
other months, while the intensity of +CG flashes has an obvious trough in August. The
discharge intensity and the proportion of +CG flashes have similar trends, with a higher
proportion and stronger discharge intensity in winter and a lower proportion and weaker
discharge intensity in summer.

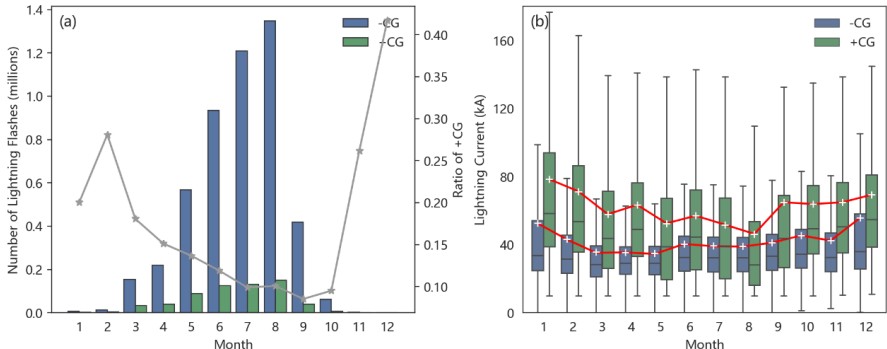

**Fig. 7. Seasonal variation of (a) frequency and (b) discharge intensity of +CG and -CG flash. The gray line**
**in (a) represents the percentage of +CG flashes, the red line in (b) represents the average current of each**
**month, and the -CG flash currents are expressed in absolute values, the same below.**
In addition, the hour-by-hour frequency and intensity variations of the two types
of CG during the day are shown in Fig. 8. The frequency of +CG and -CG flashes has
obvious fluctuation during the day. The variation trend of +CG and -CG is consistent,
and the active period is mainly concentrated in the late afternoon to midnight. Most of
the summer thunderstorms in China derive from thermal effects. The solar zenith angle
rises after noon, radiative heating keeps enhancing, thermal conditions become
abundant and conducive to the development of convection, and lightning activity is
gradually activated. With the accumulation of water vapor, lightning frequency peaks
at 17:00. After nightfall, following the weakening of thermal conditions and unstable
energy, the lightning decreases continuously and drops to a low at 11:00 the next day.
The proportion of +CG flashes is inversely correlated with the amount of CG flashes,
but the phase difference lags 1-2 h. The ratio of +CG flashes is highest at 10:00, above




15%, and lowest at 15:00, only 2%, with two sub-peaks at 22:00 and 6:00, respectively.
The current distribution and average of +CG and -CG flashes are shown in Fig.
8(b). The daily variation is not apparent. The current intensity is slightly lower from
12:00 to 19:00 in the afternoon, and the lowest point is at 14:00, while the other periods
are relatively stable.

**Fig. 8. Daily variation of +CG and -CG (a) flash frequency and (b) discharge intensity.**
3.2.2 Comparison of the spatial distribution of +CG and -CG

The geography of China is complex, and the ratio of +CG and -CG flashes have
obvious variability in spatial distribution. Fig. 9 shows the spatial distribution of the
proportion of +CG flashes. In order to reduce the anomalous values, the grid with less
than 10 CG flashes accumulated in 6 years is not included and is indicated in gray. As
shown, these grids are mainly distributed in the central, western, and northern parts of
the Tibetan and the western and southern parts of the Xinjian. The proportion of +CG
flashes in Region-I, where the density of CG flashes is the highest, is low, less than
10%. The other three regions have a higher proportion of +CG, especially the North
China Plain and its adjacent Inner Mongolia and parts of Region-III, where the +CG
ratio is up to 30-40%. Overall, lower CG density mostly means higher +CG proportion,




and high latitude corresponds to high +CG proportion.

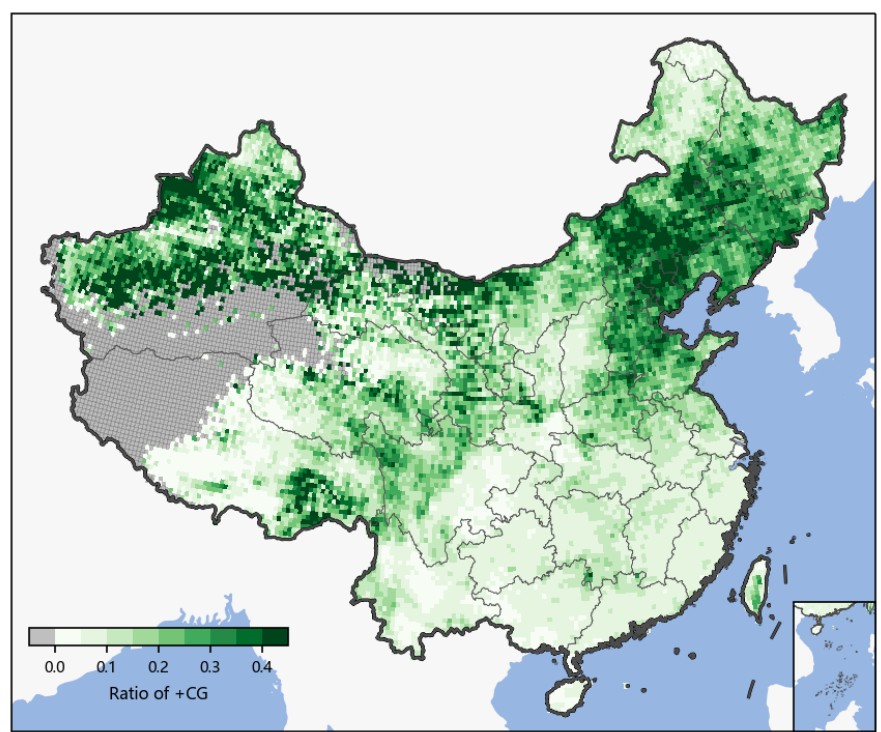


**Fig. 9. Distribution of the ratio of +CG flashes in China. The gray grids have a flash number of less than 10**

**in 6 years and thus are not calculated.**

The proportion of +CG flashes in different altitude layers is calculated, as shown
by the gray line in Fig. 10. Below 4500 m altitude, the proportion rises with the increase
of the altitude, from 7% to 15%. There is a sub-peak at 1500 m, caused by the high ratio
region of +CG flashes in Xinjiang and Inner Mongolia. Above 4500 m altitude, which
is mainly the uninhabited area in the western and northern Tibetan Plateau, the
percentage of +CG decreases rapidly. Only 91 CG flashes occurred above 6000 m
altitude in 6 years, so they are not included in the statistics.
The current distribution of +CG and -CG at different altitudes are represented by
the box plot in Fig. 10. The distribution of lightning current shrinks with increasing
altitude. A fascinating phenomenon is that the average current of -CG flashes is slightly
positively correlated with altitude, but +CG flashes are negatively correlated with
altitude. The opposite trend of the two causes a large difference in the discharge
intensity at low altitudes, but their discharge intensity is close to the same at high
altitudes.


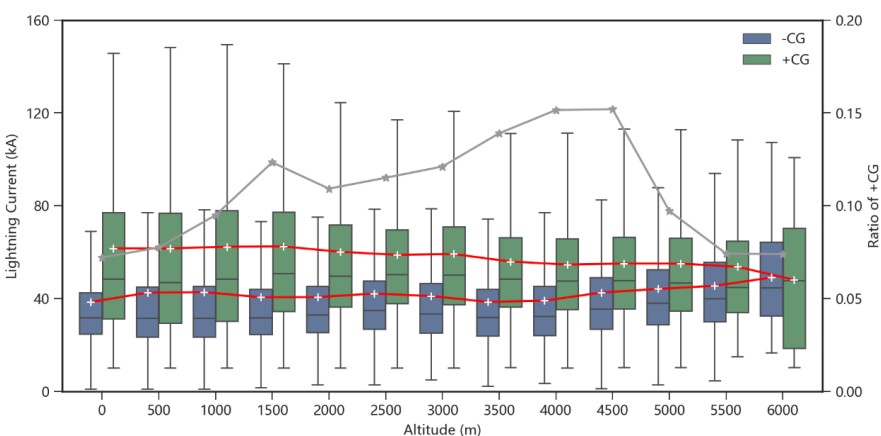


**Fig. 10. Relationship between the distribution and altitude of +CG and -CG.**

Part of the reason for the complexity of lightning activity in China comes from the
Tibetan Plateau, the "third pole" of the Earth, where the charge structure of
thunderstorm clouds has some special characteristics due to the high-altitude ground
surface. Qie et al. (2002) analyzed the lightning characteristics of thunderstorms on the
Tibetan Plateau at an altitude of 1600 m using the lightning location system installed in
Pingliang, Gansu, and found that the proportion of +CG flashes is 16% on average,
which is higher than that of conventional thunderstorms. Zhang et al. (2007) studied
the charge structure of 30 thunderstorms in Nagqu at 4500 m altitude and concluded
that although the plateau thunderstorms have a relatively strong lower positive charge
region, the lack of a negative charge region that excites the positive charge region to
discharge to the ground makes it difficult for +CG to occur. On the contrary, in our
study, the proportion of +CG flashes in Nagqu is higher than the average, see Fig. 9.
You et al. (2019) analyzed the characteristics of lightning activity in the 18-36°N Asia-
Pacific region using lightning imager data from the TRMM satellite. The statistical
results showed the discharge intensity of lightning in the Tibetan Plateau is smaller than
in other regions at the same latitude. The results of different studies vary, and there is
still no clear and unified understanding of the activity characteristics of thunderstorms
on the plateau. Further detailed analysis is in demand in combination with topographic
and climatic features.
4. Conclusion

China is mainly located in the temperate and subtropical zones, under the
combined influence of cold and warm monsoon, the interaction of land and sea, and the
undulating terrain. The above factors lead to frequent convective weather combined
with abundant lightning activities. Compared with the previous studies on China's
nationwide lightning characteristics based on satellite or thunderstorm day observation





from meteorological stations, the lightning location system used in this paper has
relatively higher DE and smaller location error for CG lightning.
ADTD and 3D-LLS are two of the three nationwide lightning location systems in
China, capable of CG lightning detection. 3D-LLS has an additional function of IC
detection. By comparing the observations of the two systems in 2020, it is found that
the overall DE of 3D-LLS is about twice that of ADTD. The DE distribution of ADTD
is relative uniform, while 3D-LLS has extreme high DE in some parts of Yunnan,
Sichuan, Tibet, Inner Mongolia, Hebei, Jilin, etc., where the sites are dense and low DE
in central Tibet, western Xinjiang, eastern Qinghai, eastern Heilongjiang, etc., where
the sites are scarce. After matching the same radiation sources detected by the two
networks, it is found that their detection time difference for the same return stroke is no
more than 10 μs, the distance difference is mostly within 2 km, and the current peaks
are almost equal. However, the DE of ADTD and 3D-LLS corresponds to only 24.5%
and 50.5% of the optical observation in the 3 km radius around the Canton Tower in
Guangdong. Besides, 3D-LLS has a high probability of misjudging IC flashes as +CG
flashes, leading to the ratio of +CG flashes up to 21.4%, much higher than the ratio
(8.5%) of ADTD. The proportion of IC flashes detected by 3D-LLS is only 1/3, which
means the DE of IC flashes is still insufficient. Both two systems need to be further
improved compared with other international developed networks.
The ADTD dataset for the past six years combined with surface height data is
utilized to analyze the CG lightning characteristics on the land of China. In general,
more in southern regions than northern regions, more in the mountains than plains at
the same latitude, more in humid regions than arid regions, and more in coastal regions
than inland regions within the same climate zone. The region with the highest CG
flashes density is the southeast coastland. The lowest density is in the northwest deserts
and basins and the east and north Tibetan Plateau. The monsoon is the main factor
affecting thunderstorms and lightning in southern and northern China, and the Tibetan
Plateau leads to the complexity of lightning activity in northwestern China and the
Qinghai-Tibet region of China. Nationally, the lightning distribution is consistent with
the precipitation on a climatic scale. According to the average CG density of provinces
and municipalities, Fujian, Zhejiang, Jiangxi, Guangdong, etc., are in the leading
position. The provinces with the lowest density are Xinjiang, Tibet, Gansu, Qinghai,
Ningxia, etc.
Due to the different mechanisms of +CG and -CG flashes, their spatial and
temporal distribution characteristics differ greatly. In terms of time distribution,
lightning activities are most active in summer (70.7%), followed by spring (19.1%) and
autumn (9.8%), and scarcest in winter (0.4%). Lightning in spring, autumn, and winter
is mainly concentrated in the southeastern coastal areas. In August, when CG lightning
is the most frequent, the proportion of +CG is the lowest, less than 10%, and its
discharge intensity is weak. In December, with the least lightning number, the
proportion of +CG rises to 40%, while the discharge intensity is stronger. Within a day,
17:00 is the peak period of the lightning frequency, and 11:00 is the trough. Similarly,



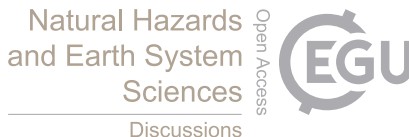

the proportion of +CG is inversely correlated with the lightning frequency, and the
discharge intensity of lightning is relatively weak in the afternoon when the lightning
is frequent. The +CG and -CG flashes ratio also show evident spatial distribution
variability. Southern China, with the highest CG density, has the lowest +CG ratio (less
than 10%). The high-latitude regions such as the North China Plain and its adjacent
parts of Inner Mongolia and northern and central Xinjiang have a 30-40% ratio. The
ratio of +CG below 4500 m is positively correlated with the altitude and decreases
rapidly after exceeding 4500 m at the western and northern Tibetan Plateau. The
discharge intensity of +CG decreases slightly with the increase of altitude, while that
of -CG increases with the altitude. The discharge intensities of the two types have a vast
difference at low altitudes and tend to be the same at high altitudes. The above rules
have also been verified by the dataset from 3D-LLS.
The lightning location system sites cannot be evenly distributed due to geographic
factors, thus bringing about errors in the lightning distribution. The observation from
Lightning Mapping Imager (LMI) on the FY-4A satellite will be used to correct the
distribution deviations by ground-based data in our following research.

## 524 Acknowledgments:

This study is supported by the Key Technologies Research and Development
Program of China (2020YFB1600103). We appreciate the Meteorological Observation
Centre of CMA and the Institute of Electrical Engineering of CAS for their data support.
We also thank the reviewers and editors for their valuable suggestions for this study.

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
