# Peer review of "Characteristics of cloud-to-ground lightning (CG) and differences between +CG and -CG strokes in China regarding China National Lightning Detection Network"

_Natural Hazards and Earth System Sciences, 2022_

## Author Comment (AC1)

**Authors reply to reviewers' comments**

Dear Anonymous Referees,

Thanks for your careful review of the manuscript. We read the reviewers' comments carefully, have considered and responded to all the reviewers' comments, and revised the manuscript accordingly. My detailed responses, including a point-by-point response to the review and a list of all relevant changes, are as follows:

**Reviewer #1: The paper analyses lightning data from two Lightning Location Systems (LLS) in China recorded during one year (2020), in a first part, and the characteristics of the CG lightning flashes by using data from one LLS recorded during 6 years (2016-2021), in a second part. The link between the two parts is not obvious since the choice of the LLS for the second part was apparently due to the dataset availability. There is no clear conclusion in the first part which could guide the choice for a set of data from an LLS to achieve the second part of the study. This first part, if it is included, must be more justified and shown as more relevant. The second part is consistent enough to make interesting the study, but many ambiguities do not allow to understand and follow the thread of the analysis to make an evaluation. The main ambiguities are related to the difference between flash and strokes which is not clearly announced, to what is provided in the datasets (flashes or strokes), to the use of two terms for the same parameter (or not, we cannot know) rate and proportion of +CG, to the current and the discharge intensity**
**The paper must be deeply revised to clarify many questions of terminology and several missing information and justification.**

**Response:** Through careful consideration, we also agree that the first part is somewhat irrelevant to the second part, so we decided to keep only the second part of the content. And following your valuable comments, the second part of the content has been extensively revised and enriched. The wording of the entire text has also been significantly modified. There are three main changes made:

a. The original system abbreviation, ADTD, has been changed to CNLDN (China National Lightning Detection Network) based on its latest official name, and a valuable site layout has been obtained, as shown in Fig. 1.

b. When analyzing the differences between +CG and -CG flashes, a more detailed statistical analysis was performed by dividing China inland into four regions according to the opinions of the second reviewer.

c. The analysis of the distribution of +CG and -CG discharge current was added.

1. The authors talk about detection efficiency (DE) in the abstract at line 22 (with values of 24.5% and 50.5%) but they do not indicate for which lightning entity it is applied: flashes or strokes? It is important to know at many steps in the study what is considered. Indeed, the DE is larger for flashes than for strokes, at least for -CG flashes. Very often in the paper, I was confronted to this problem (another example at line 26 in the abstract, another at line 103). A complete review (text, figures) is necessary for the authors to clear up any ambiguity. For the strategy of the study, it seems the DE for ATDT is only 24.5 %, it is very low for an LLS and for CG lightning (especially if it is applied to flashes). I think this point is not well commented for the following choice on LLS data and of course it should be interesting to well know if it was values for flashes or strokes, furthermore to make comparison by discussion with other LLS systems in the world.

   At line 132, I read "Time of occurrence, latitude, longitude, current peak value, number of located stations, (type of lightning) for each flash was obtained." Does it mean the strokes were not available? In figure 3 caption, it is indicated "strokes". Clarify the whole paper with that.

**Response:** Actually, the ambiguity of flashes or strokes did cause problems in the presentation of the text. In fact, as stated in lines 100-103, the lightning data used in this study are flashes grouped from strokes. So the detection efficiency was all of the flashes. The entire text has been checked and corrected. In fact, after removing the first part, the remaining contents have few descriptions of the detection efficiency.

The flash DE of 24.5% is indeed fairly low for a system working at the VLF band. But this percentage is likely to be underestimated because, in our last vision, only lightning detected by five or more stations was retained. However, in the new version, we did not perform such strict pre-screening of the data.

In addition, CNLDN (ADTD) is the only nationally deployed network in the China meteorological service and has not been evaluated on a national scale. Although the detection efficiency has relatively poor performance compared with other international developed networks, the analysis of the overall distribution trend of lightning and the comparison of +CG and -CG in China is still of scientific significance.

2. The problems of terminology can be grouped in a same comment, many times I noted fluctuant terms for an apparently same parameter:
   (i)     for the +CG two words are used, ratio and proportion. The first example is at line 34 where "ratio" is used and a value is given in %. We can logically understand it is the ratio between +CG and -CG (but strokes or flashes we do not know). But, in the paper "proportion" is also used (first at line 372) and logically the proportion is calculated by +CG / CG and not +CG / -CG. It is necessary to use the same word (and the same parameter) everywhere to understand. They have to make a clear choice.

**Response:** Thanks kindly for your comments. The entire text and the labels on the figures have been corrected. We consistently use "ratio" to indicate the ratio between the two types of lightning flash (e.g., +CG / -CG) and "proportion" to indicate the proportion of a particular type of lightning flash in the total lightning flash (e.g., +CG / CG).

(ii)    the second word to be corrected (and clarified) is for the current. The parameter provided by the LLS is the "peak current" for the strokes identified. Thus, the authors could use this word "peak current" (and to say for which stroke it is used). In a flash there are often several strokes and therefore several peak current values. Which one is used when the authors consider the flashes in the figures. Example at line 35: "The discharge intensity of +CG and -CG on the Tibetan Plateau is approximate, while the +CG always has a larger current than -CG on the plains" two words for the peak current and CG? which stroke is considered in the flash?

**Response:** Thanks kindly for your comments. The additional supplement has been added at line 103 "the first detected stroke representing the entire flash" .

3.  About the first part of the paper, the comparison of two systems in China. It can be interesting for the community if general rules are pointed out. The first remark which challenged me is at line 95: "In reviewing the literature, comparative evaluation of these two networks is lacking and mainly aimed at localized areas." For who it is lacking? I am not sure these two specific networks (distribution and location of sensors, type of sensors, treatment of data) allow to generalize some results, and do the author reach information with a certain degree of universality in the study?

**Response:** Thank you for your comments, the first part of the content has been removed.

4.  At line there is a sentence about the selection of data: "As 3D-LLS only retained lightning detected by five or more stations simultaneously this year, accordingly, this study did the same for ADTD data" Is it relevant? It means some flashes can be eliminated on one sensor and not on the other because the distribution of sensors is different for a system and for the other? I do not understand this criterion, it seems not relevant at all. Furthermore, is it applied for the second part of the paper when the characteristics of the CG lightning activity is analyzed for 2016-2020?

**Response:** In fact, the location results calculated by observations of five or more sites simultaneously would be more accurate but would miss many cases that were only detected by fewer sites, making the detection rate relatively low. However, we have removed the first part, which made data filtering just for the fairness of the two networks. In the second part of the analysis, the data were not filtered out by the site number. In fact, as mentioned in lines 98-99, CNLDN (ADTD) localized lightning using the TOA

method, which means that detections from three or more sensors are needed in the algorithm.

5.  For the different maps represented in the figures, an information of distance scale could be given, longitude and latitude on the edges?

**Response:** Thanks for your suggestion. All maps in the article have been marked with latitude and longitude.

6.  At line 173: "The DE difference between the two systems can be up to a hundred times." It is a huge difference! It would mean (for example) one is 5% and the other 50%! Is it significant? Is it calculated within an area large enough? For one pixel it is not significant. Does it mean the area is not covered? For which LLS is it a hundred larger?

**Response:** 3D-LLS has a blind detection area in the northwest of China and the Qinghai-Tibet Plateau, so the results are a hundred times smaller compared with the results under the coverage of CNLDN (ADTD). In fact, such a comparison is meaningless, so the part about the comparison of the two systems has been removed.

7.  Figure 2 is the distribution of flashes versus peak current values. On the vertical axis it is ambiguous to note ratio. It is a proportion. There is no flash at low values of peak current (for both polarities) for ADTD. Is it already filtered and not for the 3D-LLS?

    At line 184 why to say the distribution is the same for both systems? The values are low but in proportion the difference seems large above 59 kA?

    At line 186, "of outliers in the 0-30 kA range" why between 0 and +30 kA and not between 0 and -20 kA?

    At line 190-191: "direction is much larger than that in the horizontal direction, so a significant number of misjudgment cases appeared" At which value of height a detected source is considered to belong to a CG flash?

**Response:** Thank you for your comments. The first part of the content has been removed.

8.  Section 2.3. The references of studies given at lines 199-204 seem to use very different criterion values, probably because they do not consider the same matching, maybe strokes for some and flashes for other? It depends which systems are compared. This information is not discussed. The authors choose 1 s for the time criterion, it can be justified for flashes but it is not indicated.

It is not possible to understand the comment at lines 210 and 212 with the figure 3a. A proportion (clarify ratio in the figure) < 0.012 is not large. But the figure 3a displays the interval of time 0-50 μs, where are the other values? This distribution is difficult to analyze! Make another distribution and express the proportion in % it is easier to understand. Now in Figure 3 caption, I see "strokes"! Ratio is not clear in Fig. 3a,b

Finally, at line 227, we understand that the strokes are considered for matching. In these conditions, the Δt (delta t) cannot be as large as it is considered (0-1 s, line 210). If Fig. 3c include the common strokes from both systems, a time difference close to 1 s cannot be considered, such time intervals are not consistent with common strokes (not physically consistent). It can explain many dots largely out of the main cloud that follows a line.

**Response:** Thank you for your comments. Related content has been removed.

9. For the lines 245-248, it seems the negative CG flashes are also filtered according to the figure 2: no CG (flash or stroke) between -10 and 0 kA. It is not clear. Again, a lot of clarifications are necessary.

**Response:** Related content has been removed.

I do not see the utility of Figure 6; a ranking of the provinces is not scientific informative. The density distribution in Fig. 5 is much more informative.

**Response:** Fig. 6 in last vision has been removed.

10. Lines 374-375 and figure 7: the authors use ratio and proportion for +CG, it is not the same, ratio can be +CG/-CG or +CG/CG and proportion is +CG/CG. Clarify and use proportion (I think) in all figures and text.

**Response:** The entire text and the labels on the figures have been corrected. We consistently use "ratio" to indicate the ratio between the two types of lightning flash (e.g., +CG / -CG) and "proportion" to indicate the proportion of a particular type of lightning flash in the total lightning flash (e.g., +CG / CG).

The comment on "peak currents" at lines 382-383 is not relevant. The peak current values have to be explicit: average, median, others? And for -CG it also varies during the months between January and December.

**Response:** The boxes in Fig.4 and Fig.6 represent the distribution of the peak current of the flashes (the first stroke represents the flash). The white crosses and red lines in Fig.4 and Fig.6 represent the average peak current of each month. In the meanwhile, we also use "discharge intensity" to replace the current of the flash. The corresponding

text has also been revised.

11. For the hour-by-hour frequency and intensity variations, the time is not clear: define time CST. Normally CST is central time in US/Canada. The problem for China is the size, how many time zones and how to consider the same solar time or same conditions in solar influence for the figure 8? Figure 8: CST not defined, the curve is not defined. For panel a, the ratio values could be better clear with an interval between 0 and 0.16 to well show the variation amplitude.

**Response:** We apologize for the mistake of the time zone. In fact, the time zone used in this article is Beijing time, abbreviated as CNT (UTC+8). Fig.5, Fig.6 (Fig. 8 in the last vision), and all corresponding content have been corrected. As the whole China inland uses Beijing time consistently, so we only use one time zone for the analysis roughly, which will inevitably bring some bias to the analysis of the daily lightning variation. However, in the new version, when analyzing the features of +CG and -CG flashes, we conduct statistical analysis separately for four geographic regions, which can reduce the errors caused by using the same time zone.

In Figures 3, 4, 5, and 6, in order to make the comparison between regions more evident, we use the same y-scale range for each subplot.

At line 403, I do not see the same value for "lowest at 15:00, only 2%,": according to figure 8: the value seems to be between 0.09 and 0.1 (between 9 and 10 %). Again ratio is not clear.

**Response:** We apologize for the wrong ratio, and the real proportion is 9.6%. In the new vision, as we analyze the proportion of +CG separately in four regions, the related content has been replaced.

12. Section 3.2.2 is concerned by the ambiguity between ratio and proportion (line 411 "ratio" , line 416 "proportion"). What is plotted in Figure 9?

**Response:** We have consistently used "ratio" to indicate the ratio between the two types of lightning flash (e.g., +CG / -CG) and "proportion" to indicate the proportion of a particular type of lightning flash in the total lightning flash (e.g., +CG / CG).

The variable in Fig. 3, Fig. 5, Fig.7, and Fig.9 is proportion and has been corrected.

Lines 453-456: the sentence is not relevant. What is the idea there? LIS on TRMM could not estimate the discharge intensity (if you consider that as the peak current). LIS is an imager and the light from the flash that reaches the sensor is scattered by the cloud, the magnitude (optical) depends also on the location of the flash within the cloud. Anyway, LIS data does not provide discharge intensity, at least directly.

**Response:** Thanks for the reminder, the relevant content has been removed.

13. Conclusion. At line 476, it is indicated: "it is found that their detection time difference for the same return stroke is no more than 10 μs", is it consistent with Figure 3a? When you look at the values of the vertical axis, you do not see that.

At line 481: "the ratio of +CG flashes up to 21.4%, much higher than the ratio (8.5%) of ADTD". The values are averaged about the whole area and the whole year 2020 probably. Is the value 8.5 % consistent with Figures 7 and 8? In Figures 7 and 8, the minimum value of the ratio is between 0.09 and 0.1 (closer to 0.1). Is it the same parameter? What happens to have only 8.5 %? In both cases it is called ratio of CG+.

**Response:** The comparison of the two systems has been removed.

14. Minor comments:
- In the title "cloud" instead of "could"
- line 28: Thus,
- line 45: Therefore,
- line 144: "than for CG"

**Response:** Above errors have been corrected.

- line 157: To write "Fig. 1(a,b)" I think it is better "Fig. 1a,b"
- line 159: The colored scale (typical colors) is displayed in the figure, not useful to write that. Better to comment with maximum values reached in the figure.
- line 173: Is the sentence correctly written?
- line 179: "The peak current values of CG flashes detected by the two networks are compared (removing outliers above ±300 kA),"
- line 181: rewrite the sentence with a value. Most is vague and rephrase with lightning flash and peak current.
- line 250: "shows" It is present, the figure is in the paper.
- line 251: "Earth"

**Response:** Above contents have been removed.

- line 270: What is the resolution for the density calculation?

**Response:** All density maps have added a declare "The gird size is 0.25° ×0.25°"

- line 295: The shape of trumpet is not useful,

**Response:** The corresponding content has been deleted.

- line 345: It is a little strange to announce that now, the -CG and +CG have been already

discussed before in the paper.

**Response:** After removing the previous content, here is the first description of the difference between +CG and -CG flashes in lines 198-206. We think it makes sense to introduce the two types of CG flashes before discussing the difference between them.

- line 350: reference is not correct.

**Response:** The corresponding reference has been deleted.

- line 447: check the reference, the name must be written, not the first name for Qie I suppose.

**Reponse:** Thanks for your detailed suggestions. We have checked and corrected all the references.

- line 450: What does "that excites the positive charge region" mean? Why not the reverse? Which charge region excites the other? It is not really like that in the cloud physics. The maximum electric field region is generally between both regions (when they are extended).

**Response:** The corresponding content has been deleted. In new vision, we focused on the presentation of data results rather than the analysis of mechanisms.

---

## Author Comment (AC2)

**Authors reply to reviewers' comments**

Dear Anonymous Referees,

Thanks for your careful review of the manuscript. We read the reviewers' comments carefully, have considered and responded to all the reviewers' comments, and revised the manuscript accordingly. My detailed responses, including a point-by-point response to the review and a list of all relevant changes, are as follows:

**Reviewer #2:** ADTD and 3D-LLS are both nationwide Lightning Location Systems (LLSs) in China. However, up to now, the performance of the two LLSs in the whole country is not clear yet. This manuscript compared the records of the two LLSs during 2020 in the first part, and then analyzed the temporal and spatial CG distributions as well as the difference between +CG and -CG over China based on the ADTD dataset during 2016-2021 in the second part. It must be pointed out that the analysis and conclusions in the second part are meaningful only when the performance parameters of ADTD are available. However, the current analysis in the first part is definitely not enough to provide reliable performance evaluation of ADTD. It is suggested that the authors make a comparative analysis between ADTD dataset and contemporaneous data from other LLSs with known performance evaluation parameters (such as WWLLN), to get a general understanding of the overall performance of ADTD in China.

**Response:** Through careful consideration, we also agree that the first part is somewhat irrelevant to the second part, so we decided to keep only the second part of the content. And following your valuable comments, the second part of the content has been extensively revised and enriched. The wording of the entire text has also been significantly modified. There are three main changes made:

a. The original system abbreviation, ADTD, has been changed to CNLDN (China National Lightning Detection Network) based on its latest official name, and a valuable site layout has been obtained, as shown in Fig. 1.

b. When analyzing the differences between +CG and -CG flashes, a more detailed statistical analysis was performed by dividing China inland into four regions, in your valuable opinion.

c. The analysis of the distribution of +CG and -CG discharge current was added.

Major revisions are required before the acceptance of this manuscript. Major issues are listed in the following:

1. This manuscript is mainly divided into two parts, but the connection between the two parts is not tight. The support of the first part to the second part is weak. Some problems

exist in the comparison between the two LLSs in the first part.

a) Lines 13, 139, and 470: The authors declared that the CG flash detection efficiency (DE) of 3D-LLS is twice that of ADTD, because the total CG records of 3D-LLS is about twice that of ADTD in 2020. However, the authors also stated that many +CG records of 3D-LLS were misclassified from IC events. Furthermore, the comparison is limited to those records detected by 5 or more sensors and the sensors distribution of 3D-LLS network is not uniform. Therefore, such a conclusion is debatable.

b) Line 147: The authors mentioned that the CG flash density in the vicinity of the Canton Tower was 20/km^2/year. Is this an average value for several years or just for 2020? Does the term "DE" at line 153 refer to "CG flash DE"? If so, a CG flash DE of 24.5% is quite poor for a modern LLS. Considering that the sensors of 3D-LLS have already been densely distributed in southern and eastern China (see Line 117), the CG flash DE of 3D-LLS (50.5%) is also far from good. In addition, though TOLOG data can provide very good ground truth, but the comparison with the ground truth from a certain station can only give the performance parameters of a LLS in a local region, not in the nationwide area. Hence, it is necessary to conduct an evaluation for a large range combining with other data.

c) Line 172: The authors mentioned that the DE difference between the two systems can be up to a hundred times. It could also be found in Figure 1 that the DE difference between the two systems exceeds at least 10 times in many areas in Sichuan, Neimeng, Jilin, Shandong and some other provinces. This indicated that the DE of ADTD in those areas is no more than 10%, even if the DE of 3D-LLS is speculated as good as 100%. Such low DE will seriously affect the reliability of the analysis results in the second part. In addition, If the sensors distribution of 3D-LLS network is very uneven and the ADTD network is relatively uniform, it will be meaningless to compare the two LLSs in the whole nationwide region, because such comparison can't lead to a quantitative and reasonable performance evaluation results of ADTD in the nationwide area.

d) None of the sensor distribution map of the two LLSs was presented in this manuscript, which does pose a great obstacle to the understanding of the analysis results. It is suggested that the authors should not compare the two LLSs in the whole country, but to choose some certain areas where both LLSs have good sensors distribution to conduct the comparison and achieve a more reliable result. In addition, it is recommended that the authors further provide more information about the network distribution of the two LLSs as detailed as possible, such as the number of the sensors and the average baseline length in each province.

e) Line 198: The authors stated that "Comparing the detection results for the same radiation source is necessary for valuing the difference between the two networks". It seemed that the authors used a time difference threshold of 1 s for matching common CG strokes detected by both LLSs in 2.3. It can be expected that such a rough standard

will lead to a large number of mismatches.

f) Line 218: "The two networks have different criteria for grouping flashes, and if there was a missed stroke in a lightning flash and they did not use the same stroke to represent the flash, the same lightning flash could not be matched in this case, leading to the low matching ratio" is confusing. If so, why not match stroke, but flash?

**Response:** Thank you very much for your serious and meaningful comments above. The first part has been removed in the opinion of the first reviewer.
.
2. In 3.1, the authors divided China into four major regions according to geographical and climatic factors. Should these four regions be analyzed separately when analyzing the differences between +CG and -CG in 3.2?

**Response:** This suggestion helps us a lot. In the new vision of the text, we conduct statistical analysis separately for four geographic regions, which can also reduce the errors caused by using the same time zone. Also by the dividition, some interesting new findings are obtained.

3. In Figure 7b, the average peak current of +CG strokes (It is no doubt that the authors should recheck and clarify the words "flash" and "stroke" in this manuscript to avoid confusion) in summer is significantly lower than that in other seasons. Especially in August, it seemed that the median value of peak current of +CG strokes was even lower than that of the -CG strokes. Does this imply that the summer +CG dataset used in this manuscript was seriously contaminated by IC events?

**Response:** We have supplied in lines 100-103, the lightning data used in this study are flashes grouped from strokes, and the first detected stroke represents the entire flash.

In our study, +CG flashes with currents less than 10 kA are not removed to make sure the comparison between +CG and -CG fair. However, in order to prove the reliability of our conclusions, we have drawn a comparison after removing +CG flashes with peak current less than 10 kA. It can be seen from the below figures that even after removing the weak +CG flashes, the average current of +CG flashes is still lower than -CG flashes in August, which is an interesting finding of this study.

[Figure]

**Figure a. Monthly variation of the peak current distribution of the +CG and -CG flash in Southern China. The +CG flashes with peak current lower than 10 kA have been removed.**

[Figure]

**Figure b. Monthly variation of the peak current distribution of the +CG and -CG flash in Southern China. The +CG flashes with peak current lower than 10 kA are kept.**

---

## Author Response (AR2)

**Authors reply to reviewers' comments**

Dear Anonymous Referees,

Thanks for your careful review of the manuscript. We read the reviewers' comments carefully, have considered and responded to all the reviewers' comments, and revised the manuscript accordingly. My detailed responses, including a point-by-point response to the review and a list of all relevant changes, are as follows:

**Reviewer #1:**

The authors stated that: "+CG flashes with currents less than 10 kA are not removed to make sure the comparison between +CG and -CG fair". However, a lots of studies showed that most positive discharges with peak current below 10 kA are cloud discharges. So, this should be taken to mitigate such misclassification, as to improve the quality of the CG dataset used in this paper.

**Response:** In fact, in our initial version of the submission, we filtered out +CG with peak currents below 10 kA. However, the third reviewer raised concerns that this filtering could lead to an overestimation of the average peak current for +CG flashes, resulting in unfair comparisons between +CG and -CG in Figures 4, 6, and 9. Therefore, in this revised version, we reprocessed the data and did not filter out +CG flashes with peak current below 10 kA.

2. Line 103, the authors stated that they used the first detected stroke to represent the entire flash. However, for comparing the peak current of different multi-stroke CG flashes, it is not enough to consider only the first detected stroke.

**Response:** We appreciate your insightful suggestion, and upon thorough deliberation among the authors, we concur that relying solely on the first lightning stroke to represent the entirety of a lightning flash could potentially introduce bias to the conclusions drawn. Therefore, the entire text has been reprocessed, shifting from analyzing lightning flashes to analyzing lightning strokes. Consequently, the focus is no longer solely on the location and current of the first stroke, but rather on the locations and currents of all lightning strokes.

3. Line 109, it is suggested to use the term "number of reporting sensors" instead of "the number of triggered stations".

**Response:** Thank you for your suggestion. The corresponding content has been modified, see line 126.

**Reviewer #2:**

The second version of the paper is more relevant in a generally speaking. The lightning data are analyzed in several aspects for the whole Chinese territory. It is a complex task to present the variability according to many parameters. However, some comments are not correct according to the figures. The terminology is still ambiguous and unusual. English language could be improved. The authors have improved the paper but it needs a revision, more than minor according to the number of corrections to do.

This article has three major changes. Firstly, based on the feedback from the first reviewer, the statistics on lightning flash have been modified to statistics on lightning strokes. Secondly, due to the long review period, new data for another year has been obtained, expanding the data collection period to seven years (2016-2022). As a result, the relevant statistical values have undergone slight fluctuations, and the figures have been adjusted accordingly. However, the main conclusions remain essentially unchanged. Furthermore, we have also improved the comments according to the figures to enhance the reliability of the results.

1. My comments about terminology in the first review are partially taken into account. The main which is still to be clarified is the "peak current" for the strokes and consequently for the flashes since the flashes take the peak current of the first stroke. The peak current is the characteristic provided by the detection systems. The authors employ often "intensity" or "current". It is necessary to standardize in the text and as it is made in other works, with "peak current", it will be clearer.

**Response:** Thank you for your suggestion. The phrase "discharge intensity" has been replaced with "peak current" throughout the paper.

On the other hand, for the proportion and ratio concerning the +CG flashes, in the abstract I read two different terms: line 21 "+CG proportion" and line 29 "the +CG ratio". By using proportion, it is clear; by using ratio it is ambiguous! Is it +CG/-CG ratio or +CG/CG ratio? Again, it is necessary to use the same term (in the whole paper) to avoid ambiguity. When it is expressed in % it is clearly a proportion, when it is decimal number it is a ratio and it is ambiguous. I recommend to use the term "proportion" and the % for the values.

**Response:** Through a thorough text search, we found that there is only one instance of the term "ratio" remaining in the abstract. We sincerely apologize for this oversight even after the first round of revisions. We have now replaced the term "ratio" with "proportion" in the abstract. In fact, the intended meaning of the paper is the "+CG/all CG" ratio, so using the term "ratio" was incorrect

2. In a lot of comments, the authors use region names which are not well identified by the reader. For many comments, the geographical location (longitude, latitude) could help to understand. Maybe hen a region is cited and commented, a reference could be made also to Figure 1 with the initials? An example at lines 325-326.

**Response:** The geographical abbreviations in Fig. 1 have been replaced with their full

names, and the geographical names unfriendly for non-Chinese readers mentioned in the text have been labeled in Fig. 1 using the alphabetical labels (a, b, c, d, etc.).

3. The efficiency is commented in the conclusion (not before in section 2) as "relatively high" but it is not quantified. First, it could be made in section 2 with values or range of values according to the region. Secondly, in the first version of the paper it was given at 24.5 % for the national system, it is not high. It is not clear at that point.

**Response:** Indeed, the detection efficiency of CNLDN is relatively low compared to other internationally advanced systems, such as NLDN of America. Besides, there has been no research conducted to evaluate the detection efficiency across all regions covered by CNLDN.

Nevertheless, CNLDN remains the sole operational system employed by the Chinese meteorological department, serving as the most extensively deployed national lightning detection system in China. It should be noted that the term "relatively high" in the original conclusion refers to a comparison with other detection systems within China; however, this expression is imprecise. Therefore, it has been modified to "This paper utilizes the dataset from a ground-based lightning location system, CNLDN, which serves as the most extensively deployed national lightning detection system in China, to analyze the CG lightning characteristics in China over the past seven years." at lines 404-407.

In addition, the current research progresses on the local assessment of CNLDN are added in the introduction at lines 87-100.

4. The term of complexity is often used in the paper. It is not clear what is signified by complexity, for example in the conclusion, at line 374. The complexity is a little vague. I think the interpretation have to be more accurate when it is possible.

**Response:** The sentences related to the term "complexity" in the text have all been modified. See lines 27,337,392-395,416-418.

5. The data used correspond with lightning flashes over 6 years. The total number of flashes could be given somewhere (in the abstract and in the text with more details according to the different regions. A table could summarize the activity by year and by each of the four regions, it would be welcome. I would say even it is necessary.

**Response:** Tab.1 has been added to the text displaying the statistics on the annual average numbers of return strokes, stroke densities, and peak current values of the four regions.

6. At lines 241-242 and Figure 3, the comment has to be revised, it is not true for all regions. For example, in region I (the most active), the +CG proportion is the lowest in months September and October! In other regions, it is also around September that it is the lowest, but in region II it is also low in March. It has to be revised in all the paper I think. At line 243, the value is lower than 10 % also, it has to be revised.

**Response:** After reprocessing the data, the corresponding descriptions in the document

have also been modified. Please refer to the highlighted sections at lines 273-280.

7. Figure 5: Maybe the scale has to be adapted in each panel? The comment at line 270 is not relevant, the lightning activity at midnight is not so strong compared to the beginning of the day (01h 00) for most regions. The sentence has to be revised. The maximum is around 16h for most regions. The sentence at line 279 is wrong, it is not true for regions II and III (both parts of the sentence). Furthermore, it is not consistent with the sentence at line 278.

**Response:** The scale has been adapted in each panel for Fig. 5. But after carefully checking, the maximum is around 15:00 CST in Region-I and Region-II in the east of China and 1-2 hours later in Region-III and Region-IV in the west of China as illustrated in the text. The other modifications have been highlighted in red in the text.

8. The comments of Figure 6 could be clearer and more correct (lines 292-300). For example, at line 295 "with a more intense change in -CG than in +CG.", I see the opposite! This part has to be more fluent for the language. At line 300 for the disparity, it is also large before noon, correct during noon by around noon maybe?

**Response:** Based on your feedback, we have made extensive revisions to the description of Fig.6 in this paragraph. See lines 330-341.

9. For the average peak current in Figure 8, some values are surprising (around 200 kA) for such a long period (6 years). Is it correct? Is it consistent with Figure 9 which does not show such large values in the distribution?

**Response:** Through meticulous examination of the data, we have discovered that over a span of seven years, there were indeed instances of strokes with exceptionally high current values. Approximately 5.1% of the stokes with peak currents exceeding ±100kA, while 0.7% of the strokes surpass ±200kA. The distribution range of peak current of +CG stroke is also consistent with the results inferred from the radiation electric field by *Nag, Amitabh, Vladimir A. Rakov, and Kenneth L. Cummins. "Positive lightning peak currents reported by the US National Lightning Detection Network."IEEE transactions on electromagnetic compatibility 56.2 (2013): 404-412.*

This observation aligns with Fig. 9, as the box plots depicted in Fig. 9 have not displayed the outlier values. Hence, there is no contradiction between these findings.

At lines 242, only 91 CG flashes above 6000 m: is it a problem of detection? A small area at this altitude? Usually mountains are favorable for lightning activity.

**Response:** In fact, according to the description provided in lines 115-116, CNLDN indeed has some blinds areas in the desert regions of Xinjiang and the uninhabited high-altitude areas of Tibet Minor comments, which will somehow lead to lower observed results. So,"91 CG flashes occurred above 6000 m" has been revised to "91 CG flashes were observed above 6000 m".

Although mountains are favorable for lightning activity, the extremely high-altitude regions above 6000m are typically covered by year-round ice and snow, resulting in a

minimal occurrence of lightning events. The aforementioned conclusion can be supported by satellite-based lightning observations below.

[Figure]

**(Lightning observations from1994 to 2013 by LIS on TRMM)**

At line 345, it is not true for the -CG flashes, it does not decrease with altitude (only true for +CG flashes). At line 350, it is not clear since the lightning activity is low in this plateau. More detailed explanation about "complexity" is necessary.

**Response:** Thanks for your remind. It has been revised to "The distribution of +CG peak current narrows with increasing altitude."

At line 350, it is not clear since the lightning activity is low in this plateau. More detailed explanation about "complexity" is necessary.

**Response:** The statement in the original text has been changed to "The Tibetan Plateau is primarily responsible for the intricate lightning activity versus altitude over China. As the "third pole" of the Earth, the charge structure of thunderstorm clouds on the Tibetan Plateau always has some special characteristics due to the high-altitude ground surface. Furthermore, its influence on the uplift and obstruction of water vapor can also affect the climatic characteristics of other regions.". See lines 392-397.

*Minor comments:*

- line 18: CNT is never defined in the paper. It is necessary to do it in both abstract once and in the text at the first use.

**Response:** Thank you for your reminder. This article has changed all the CNT in the text to CST (China Standard Time) and given the full name at the first appearance in the abstract and the main text.

- line 37: what is meso-small? Is it used in the literature?

**Response:** After consulting the literature, We found that this expression is indeed uncommon, so we changed it to "mesoscale or small scale" in the article.

- line 38: lightning is usually associated with cumulonimbus, stratus is not really associated.

- line 39: nuclear explosions are not frequent (hopefully!), is it necessary to indicate it?

- line 52: replace "could" by "can"

- line 154: "grid"

**Response:** Thank you for your reminder. The relevant descriptions have been revised according to the above four comments.

- line 202: "a larger spatial scale" it is not clear where the scale is considered, in the cloud?

**Response:** It has been deleted.

- line 208: space. Often the space has to be added.

**Response:** Thanks for the reminder, we have done a full-text index and added the missing spaces.

- line 252: "peak current"

- line 266: "The red lines represent" there are two red lines.

**Response:** Thank you for your reminder. The relevant descriptions have been revised according to the above two comments.

- line 268: the sentence has to be revised. In Fig. 5 the peak current is not plotted but the proportion of +CG.

**Response:** The sentence has been revised to "Fig.5 illustrates the hour-by-hour frequency and proportion variations of CG flashes throughout the day."

- line 291: "peak current". Rephrase with "The hourly distribution of the peak current value and its average are shown in Figure 6 for +CG and -CG flashes."

**Response:** This statement does make the description clearer and more reasonable, and it has been modified according to your comments.

- line 295: "at noon"

- line 303: "for each hour and not each month..

- line 312: space before % (in other parts also).

- line 321: grid (and other figure captions).

**Response:** Thank you for your reminder. The relevant descriptions have been revised

according to the above four comments.

- line 328: is it necessary to say "in terms of temporal and spatial scales"? If it corresponds it is enough.

**Response:** I think it's important to emphasize this, because this article is discussed separately in time and space, and we can all reach a consistent conclusion. Both in time and space scales, the two have good consistency.

- line 378: Rephrase "due to their different mechanisms" with due to different storm structures" if you agree?

**Response:** I agree that this change has a better effect.

- line 393: delete "midnight" after checking. "drops".

**Response:** Modifications have been made accordingly.

- line 402: Rephrase "The +CG proportion exhibits significant spatial variability."

**Response:** It has been revised to "The distribution of the +CG stroke proportion exhibits significant spatial variability."

**Reviewer #3:**

In this manuscript the authors analyze the characteristics of cloud-to-ground lightning in China over a 6-year period as observed by the Chinese National Lightning Detection Network (CNLDN). The current manuscript has undergone previously already a round of peer-review. Before this manuscript could be accepted for publication the following points should be taken into account:

This article has three two changes. Firstly, based on the feedback from the first reviewer, the statistics on lightning flash have been modified to statistics on lightning strokes. Secondly, due to the long review period, new data for another year has been obtained, expanding the data collection period to seven years (2016-2022). As a result, the relevant statistical values have undergone slight fluctuations, and the figures have been adjusted accordingly. However, the main conclusions remain essentially unchanged. Additionally, several citations of relevant research progress have been added.

- Acronyms: CNLDN (it is common practice not to use acronyms in the title), CNT (in abstract, and later in the text), TRMM (introduction at L79) should be defined before the acronym is used.

**Response:** The title has replaced CNLDN with its full name, and other abbreviations in the text have also been marked with the full name at their first occurrence.

- Throughout the text 'current peak' is used. In stead, 'peak current' is what is commonly used in scientific journals.

**Response:** The phrase 'current peak' in the full text has been revised.

- Introduction L46: what do you mean with 'scientific protection (...)'?

**Response:** It has been revised to "Therefore, the timely and accurate monitoring of lightning serves as an effective approach for the development of lightning science and scientifically mitigating the hazards of lightning strikes."

- Introduction L51/52: only references to some LF networks are listed, without any reference to VLF networks. Some extra references to, e.g., GLD360 & WWLLN, should be included since the authors speak about the advantages of ionospheric reflections of the radiation emitted by discharges.

**Response:** The relevant content has been added to lines 58-62.

- Introduction L59-61: it would be good to include most recent references to the networks NLDN, LASA, and EUCLID.

**Response:** The relevant content has been added to lines 58-62.

- Introduction L79: rephrase 'was no longer updated'

**Response:** It has been revised to "discontinued updates"

- Introduction L80: 'detection rate'. Do you mean 'detection efficiency'?

**Response:** Thanks. The "detection rate" has been replaced by "detection efficiency".

- Introduction L89-L91: rephrase last sentence of the introduction.

**Response:** The last sentence has been rephrased as "Furthermore, China's extensive geographical expanse, spanning a wide range of latitudes and longitudes, coupled with its intricate topography, provides a unique opportunity for investigating the correlation between lightning occurrences and geographic factors."

- Sect. 2 L100–105: 1) what about the detection of IC pulses? The focus of this paper is not on IC, however, if IC pulses are detected, it would be good to mention it in the text.

**Response:** This study collected data spanning seven years; however, the detection capability for IC pulses was only added to CNLDN in half of 2021 and 2022. After conducting statistical analysis, it was found that the detection efficiency of IC was relatively lower. Therefore, this paper does not include any relevant descriptions regarding IC detection.

2) This manuscript discusses, amongst others, the spatial distribution of CG flashes. It is therefore a necessity to state the flash detection efficiency (DE) and location accuracy (LA) of CNLDN and add references to publications thereof. Otherwise, the reader has no idea about the quality of the network, and hence, the quality of the results in this study.

**Response:** Indeed there has been no research conducted to evaluate the detection efficiency across all regions covered by CNLDN. The current research progresses on the local assessment of CNLDN are added in the introduction at lines 87-100.

- Sect. 2 L109: ''triggered stations'. Do you mean 'participating stations'? There is an important difference.

**Response:** According to the suggestions provided by the first reviewer, the " triggered stations " has been changed to " reporting sensors".

- Fig.1: what is the purpose of the inset Figure in the bottom right corner? At first glance, it does not add anything and could be removed.
**Response:** The inset map in the corner is a supplementary depiction of the nine-dash line to ensure the integrity of China's territorial boundaries. Generally, Chinese maps that comply with the verification of the Ministry of Natural Resources of China require the inclusion of the nine-dash line and the South China Sea region. As shown in the following officially released figure.

[Figure]

- Sect. 3.1.: I would advice to include mean/median values of the flash density per region. This is not included at the moment.

**Response:** Tab.1 has been added to the text including the annual average numbers of return strokes, stroke densities, and peak current values of the four regions.

- Sect. 3.1: it is better to use for the unit of flash densities 'fl km-2 yr-1', where 'fl' stands for flashes

**Response:** Your suggestion is good, but the manuscript has already been modified based on the first reviewer's recommendation to change the focus from statistics on lightning flash to statistics on lightning return strokes.

- Sect. 3.1, Fig. 2, L154: 1) 'gird' → 'grid',

**Response:** The above-mentioned error has been corrected.

2) why 0.25x0.25 degree grid is used? It would be good to justify this size. The authors could, e.g., refer to Diendorfer (2008, some comments on the achievable accuracy of local ground flashes density values, Proc. 29th Int. Conf. On Lightning Protection, Uppsala, Sweden), who demonstrated that in order to obtain an uncertainty of less than 20% at 90% confidence level, a grid size has to be chosen in such a way that the dimensions of each cell and the number of years considered both comply with the minimum requirements of the following equation, following the Poisson distribution of the law of rare events: Ng x Tobs x Acell >= 80. In your case the grid cell area Acell is about 25x25km2, Tobs=6 years. Hence, Ng >= 80 / (6x625) = 0.02 fl km-2 yr-1. It follows that your chosen grid size is ok to work with.

**Response:** Thank you for your suggestion. We have incorporated the reasoning for our choice of grid size based on the literature you provided at lines 156-161 of the text.

"According to the research of Diendorfer (2008), lightning is a highly stochastic phenomenon and if we require an uncertainty of less than ± 20 % there should be more than 80 events per grid cell. Therefore, we establish the grid size as 0.25° ×0.25°, ensuring that the results within the confidence interval for all regions, except part of areas in Xinjiang and Tibet."

- Text related to Fig. 4: it would be good to include mean/median values of the absolute peak current for -CG & +CG observed in the different regions.

**Response:** The mean peak current values have been added to Tab. 1 and relevant descriptions have been added at lines 284-288.

- What I miss in this manuscript, is some connections to observations in other parts of the world. For instance, how do the spatial densities and in particular the absolute peak currents compare to observations in other parts of the world? One could compare to, e.g., EUCLID (e.g., Poelman et al., 2016, https://nhess.copernicus.org/articles/16/607/2016/) or NLDN journal papers, since those networks are similar to CNLDN and above all are one of the best documented networks in terms of DE and LA.

**Response:** Thanks for your suggestions. The relative comparisons have been added to lines 254-266, 312-314

- Similar to previous point: the authors could include some comparison to observations made by other networks in China, e.g., Xia et al. (a 6yr cloud-to-ground lightning climatology and its relationship to rainfall over Central and Eastern China, https://doi.org/10.1175/JAMC-D-15-0029.1). How well do the current results overlap with those published in the past?

**Response:** Thanks for your suggestions. The relative comparisons have been added to lines 162-164,178-179.

- Text related to Fig. 5 L291-300: what could be the reason behind the fact that, e.g., in region I the peak current decreases 'in the noon and afternoon'?

**Response:** Throughout the analysis in this paper, a consistent finding has emerged: in periods or regions with high lightning frequency, the peak current of lightning discharges tends to weaken. In the noon and afternoon, where lightning frequency is high, the peak current values are comparatively lower. However, this paper merely objectively describes the observed phenomena, and it is difficult to provide a reasonable scientific explanation based on the existing observations.

- Sect. 3.2.2: maybe this section could be moved directly after Sect. 3.1, which deals with the spatial distribution of the CG flash density?

**Response:** Thank you for your valuable suggestion. However, Section 3.2 investigates the differences between +CG and -CG, while Section 3.1 focuses on the overall distribution characteristics and their relationship with topography and climate. We would like to maintain the original structure as it is.

- General comment: throughout the text the others mention names of places, e.g., Shanxi, Shaanxi, Guizhou, … Many, if not all, non-Chinese readers will not have a clue where those exactly are located. Maybe the others could include in some way in Fig. 1 the locations of the places mentioned in the text?

**Response:** The geographical abbreviations in Fig. 1 have been replaced with their full names, and the geographical names unfriendly for non-Chinese readers mentioned in the text have been labeled in Fig. 1 using the alphabetical labels (a, b, c, d, etc.).

- Fig. 8 L327-329: last sentence should be rephrased.

**Response:** It has been revised to "Therefore, it can be concluded that a high proportion of +CG stroke typically corresponds to larger current values in terms of temporal and spatial scales."

- Fig. 8; the authors could use a similar scale. Now, the scale for |-CG| is [0,200]kA, whereas for +CG it is [0,250] kA. I would use the same scale from [0,200] kA.

**Response:** Both figures have been modified to have the same current range of 0-250kA.

- Fig. 8: what is the cause of the much higher absolute peak current in the area around 28N, 95E?

**Response:** The exceptionally strong lightning current in Mêdog County may be because of its unique geographical location (at the gap of the Himalayas Mountains), leading to distinct charge distributions within thunderstorm clouds. However, due to a lack of relevant observations and research, it cannot be confirmed at this time.

- Fig. 9: is a similar trend also found in other parts of the world? Did the others check the literature carefully?

**Response:** As the "Roof of the World", the Qinghai-Tibet Plateau is unique to China and provides a valuable opportunity for investigating the correlation between lightning occurrences and geographic factors. Currently, we have not found other research regarding the relationship between lightning and altitude.

---

## Author Response (AR3)

**Authors reply to reviewers' comments**

**Reviewer #1:**

Most of the reviewer's comments to this manuscript have been well addressed by the authors, and the quality of the revised manuscript has been improved significantly. However, the authors still retained the +CG flashes (strokes) with currents less than 10 kA as part of the CG dataset, which may lead to serious "contamination" for the CG dataset. It is suggested to further analyze the peak current distribution characteristics for +CG and -CG stokes and propose appropriate CG dataset quality control.

**Response:** In the latest round of revisions, we have considered the suggestions from two reviewers and once again excluded strokes with current less than 10 kA. As a result, there may be some fluctuations in the numerical values in Table 1 and the displayed results in the images. However, the overall conclusions remain largely unchanged.

**Reviewer #2:**

This round of revisions primarily involves two main changes. Firstly, we removed +CG strokes with peak currents less than 10 kA. Secondly, we have corrected the previous data error by changing the statistics from lightning flashes to lightning strokes. However, the detection efficiency for strokes in our system is notably low, and the ratio between strokes and flashes is approximately 1.3. Also since we have also removed part of +CG strokes, the number changes are not substantial.

1. The new version of the paper changes substantially according to the authors responses to the reviewers:

They say for example to Reviewer 2: "This article has three major changes. Firstly, based on the feedback from the first reviewer, the statistics on lightning flash have been modified to statistics on lightning strokes. Secondly, due to the long review period, new data for another year has been obtained, expanding the data collection period to seven years (2016-2022). As a result, the relevant statistical values have undergone slight fluctuations, and the figures have been adjusted accordingly."

They also say to Reviewer 1: "Therefore, the entire text has been reprocessed, shifting from analyzing lightning flashes to analyzing lightning strokes. Consequently, the focus is no longer solely on the location and current of the first stroke, but rather on the locations and currents of all lightning strokes."

However, I see the same figures in the new version of the paper: Figure 2 does not change at all, although all strokes are now considered and one more year of data is used. The number of strokes is generally in average two times that of flashes (multiplicity 2 for CG flashes, in average and for all CG flashes). All figures are identical to the previous ones, so we cannot consider them for a new review.

**Response:** We sincerely apologize for the carelessness of invoking the previous flash

data by mistake in our codes in the last round of revisions and have taken repeated measures in this round to ensure that such errors do not occur again. We have modified Figures 2-6 and 9 based on the new data, while the data invoked in Figures 7-8 are correct. The number of lightning strokes has increased compared to the previous version; however, due to the relatively low detection efficiency of CNLDN, the number of strokes does not double that of lightning flashes. Furthermore, for Figure 2, the changes are not very pronounced as the axes are logarithmic.

2. By considering the previous comments I made and the responses of the authors, I see remaining comments which the authors say to have deleted or modified.

For example, in the abstract, I see "lightning current" instead of "peak current" at lines 22, 31 and 33; then at many places within the paper.

**Response:** We conducted a comprehensive search and revision of the entire document and changed "lightning current" to "peak current".

For the proportion of the positive CG strokes (now strokes) I see in several graphs the proportion in values < 1 as a ratio and in the text in %. I think the graphs must be changed.

Anyway, if we take into account the changes described by the authors, all figures must be changed since they are exactly as in the previous version of the paper.

**Response:** The scales for Figures 3, 5, 7, and 9 have all been modified to be in percentage format, and the textual descriptions of decimals in the text have also been changed to percentages.

3. On the general speaking, English must be improved:

Some examples:

Line 2 and many places in the paper: "+CG and -CG strokes" (if you say "and" two kind of stroke)

**Response:** Sorry. I did not figure out the meaning of this suggestion.

Line 15: "monthly scale" is during one month (as at the scale of the life is along the life). They want to say "at the annual scale", it is related to Figure 3 and the same mistake is made in the caption of the figure. This kind of mistake must be checked.

**Response:** Thanks for your reminder. The corresponding contents have been modified.

Line 90: in this new comment, there is something which is not relevant and does not correspond to the result cited from the referenced work: "the detection efficiency for lightning flash and stroke in Beijing was reported to be 36.5% and 49.4%" Usually, the detection efficiency is lower for strokes because if one stroke is detected from a flash (and not other strokes of the flash) the flash is detected, it is a typical result, but we can imagine an opposite result according to the strokes detected. Anyway, the authors (Srivastava et al., 2017) do not say that, the result is not well reported. They make the difference between CG and IC flashes with these proportions. Revise that please.

**Response:** Unfortunately, we made an error in citing the data, and actually, the detection efficiency for strokes should be lower than that for lightning flashes. This has been corrected. See Line 91.

Line 132: "in China" should be better.

**Response:** Thanks for your reminder. It has been revised.

Table 1: "k" is strangely used. We can suppose it is one thousand, but it is not common. Use the power of 10 for the related parameters in the first column.

**Response:** Thank you for your reminder. The necessary changes have been made to the table. Additionally, due to the exclusion of +CG strokes with currents less than 10 kA, there have been slight alterations in the numerical values in the statistical results.

The caption must be rephrased: "Tab. 1 Statistics on the annual average numbers of return strokes, the stroke densities, and the peak current values of the four regions"

**Response:** Thank you for your suggestion. The corresponding content has been modified.

Line 166: "Region-I has the highest concentration of CG stroke, with an average density of up to 1.69 km-2 yr-1" If it is an average (over time and area) there is only one value, do not use "up to".

**Response:** Thanks for your reminder. It has been revised.

Line 186: what is "degenerate"?

**Response:** It has already been changed to " degenerative", which refers to a tropical maritime air mass that has undergone alterations in temperature, humidity, or other properties.

4. I let the authors to clarify the new figures which are the same than the previous ones in spite of the use of one more year and strokes instead of flashes.

**Response:** Table 1 and Figures 2-7 and 9 have been corrected according to the new data.